# Cryo-EM reconstructions of inhibitor-bound SMG1 kinase reveal an autoinhibitory state dependent on SMG8

Lukas M Langer, Fabien Bonneau, Yair Gat, Elena Conti*

Max Planck Institute of Biochemistry, Martinsried, Germany

**Abstract** The PI3K-related kinase (PIKK) SMG1 monitors the progression of metazoan nonsense-mediated mRNA decay (NMD) by phosphorylating the RNA helicase UPF1. Previous work has shown that the activity of SMG1 is impaired by small molecule inhibitors, is reduced by the SMG1 interactors SMG8 and SMG9, and is downregulated by the so-called SMG1 insertion domain. However, the molecular basis for this complex regulatory network has remained elusive. Here, we present cryo-electron microscopy reconstructions of human SMG1-9 and SMG1-8-9 complexes bound to either a SMG1 inhibitor or a non-hydrolyzable ATP analog at overall resolutions ranging from 2.8 to 3.6 Å. These structures reveal the basis with which a small molecule inhibitor preferentially targets SMG1 over other PIKKs. By comparison with our previously reported substrate-bound structure (Langer et al.,2020), we show that the SMG1 insertion domain can exert an autoinhibitory function by directly blocking the substrate-binding path as well as overall access to the SMG1 kinase active site. Together with biochemical analysis, our data indicate that SMG1 autoinhibition is stabilized by the presence of SMG8. Our results explain the specific inhibition of SMG1 by an ATP-competitive small molecule, provide insights into regulation of its kinase activity within the NMD pathway, and expand the understanding of PIKK regulatory mechanisms in general.

**\*For correspondence:**
conti@biochem.mpg.de

**Competing interest:** The authors declare that no competing interests exist.

## Editor's evaluation

This study uses CryoEM and biochemical studies to uncover a new and potentially important conformational off-state of a key regulatory multi-subunit protein kinase, SMG1. The study was enabled by applying a small molecule ATP-site inhibitor to capture the structure. The work will be of wide interest to the signaling and structural biology communities.

## Introduction

Nonsense-mediated mRNA decay (NMD) is a co-translational mRNA quality control pathway central to the detection and removal of mRNAs containing premature termination codons as well as to the regulation of many physiological transcripts (*Kurosaki et al., 2019*; *Karousis and Mühlemann, 2019*). In metazoans, translation termination at a premature stop codon triggers phosphorylation of the RNA helicase UPF1, which then enables the recruitment of downstream effectors, in turn resulting in the degradation of the targeted mRNA by ribonucleases (*Ohnishi et al., 2003*; *Yamashita, 2013*; *Chakrabarti et al., 2014*; *Nicholson et al., 2014*; *Okada-Katsuhata et al., 2012*; *Kashima et al., 2006*). UPF1 phosphorylation is thus a crucial point in metazoan NMD; it occurs specifically at particular Ser-containing motifs and is mediated by the SMG1 kinase (*Denning et al., 2001*; *Yamashita et al., 2001*).

SMG1 is a member of the phosphatidylinositol-3-kinase-related kinase (PIKK) family, which also includes the ATM, ATR, DNA-PKc, and mTOR kinases (*Keith and Schreiber, 1995*). Despite the

confounding name reflecting a possible evolutionary origin with a lipid kinase (*Keith and Schreiber, 1995*), and in retrospect the ability of some PIKK family members to bind inositol-6-phosphate (*Gat et al., 2019*), all active PIKKs are Ser-Thr protein kinases. These large, multidomain enzymes share an overall similar architecture: a C-terminal catalytic module (comprising the so-called FAT, kinase, and FATC domains) and an N-terminal α-solenoid (that typically serves as a protein-protein interaction module). Indeed, PIKKs bind to and are regulated by interacting proteins, with which they form large and dynamic assemblies. In addition, some PIKKs contain an autoregulatory element, particularly in the variable region connecting the kinase and FATC domains, that is known as the PIKK-regulatory domain (PRD) (*Baretić and Williams, 2014*; *Imseng et al., 2018*; *Jansma and Hopfner, 2021*). Finally, the kinase domains of PIKKs have been the target of numerous efforts in the development of specific inhibitors. Not surprisingly, given the central roles of these kinases in pathways surveilling cellular homeostasis, small molecules targeting specific PIKKs have been approved as therapeutics or are being evaluated in clinical trials (*Janku et al., 2018*; *Zhang et al., 2011*; *Durant et al., 2018*). Nevertheless, the high conservation of the PIKK kinase domain has complicated efforts to develop inhibitors specifically targeting only a selected member of this family of enzymes. Structural data rationalizing the binding specificity of such compounds to PIKKs are still sparse in general and entirely missing for the SMG1 kinase.

In the case of SMG1, the two interacting factors SMG8 and SMG9 have been linked to dysregulated NMD and neurodevelopmental disorders in humans (*Alzahrani et al., 2020*; *Shaheen et al., 2016*; *Yamashita et al., 2009*). Previous cryo-electron microscopy (cryo-EM) studies have revealed the molecular interactions that underpin the structure of the human SMG1-SMG8-SMG9 complex (*Gat et al., 2019*; *Zhu et al., 2019*) and the determinants with which it recognizes and phosphorylates UPF1 peptides with specific Leu-Ser-Gln (LSQ) motifs (*Langer et al., 2020*). SMG8 and SMG9 form an unusual G-domain heterodimer (*Gat et al., 2019*; *Langer et al., 2020*; *Li et al., 2017*). The G-domain of SMG9 binds both the α-solenoid and the catalytic module of SMG1 while the G-domain of SMG8 engages only the α-solenoid, thus rationalizing biochemical studies pointing to the crucial role of SMG9 in enabling the incorporation of SMG8 into the complex (*Yamashita et al., 2009*; *Arias-Palomo et al., 2011*). In turn, SMG8 appears to have a direct regulatory function on SMG1, as its removal or C-terminal truncation results in hyper-activation of SMG1 kinase activity (*Yamashita et al., 2009*; *Deniaud et al., 2015*; *Arias-Palomo et al., 2011*; *Zhu et al., 2019*). However, the C-terminal domain of SMG8 remains poorly defined in all available cryo-EM reconstructions (*Zhu et al., 2019*; *Gat et al., 2019*; *Langer et al., 2020*), hindering a molecular understanding of its regulatory role.

Another portion of the complex reported to downregulate SMG1 kinase activity is the so-called insertion domain, a large 1200-residue region connecting the SMG1 kinase and FATC domains. Removal of the SMG1 insertion domain causes hyper-activation of the kinase (*Deniaud et al., 2015*; *Zhu et al., 2019*), similarly to the effect reported for the PRDs of other PIKKs (*McMahon et al., 2002*; *Edinger and Thompson, 2004*; *Xiao et al., 2019*; *Mordes et al., 2008*). However, the SMG1 insertion domain shows no sequence similarity to the PRDs of other PIKKs and is remarkably larger in comparison. None of the current cryo-EM reconstructions of SMG1 or its complexes show density corresponding to the SMG1 insertion domain (*Zhu et al., 2019*; *Gat et al., 2019*; *Langer et al., 2020*), and it is thus unclear how it regulates kinase activity.

Here, we used cryo-EM and biochemical approaches to study the basis with which SMG1 is specifically inhibited by a small molecule compound (SMG1**i**), and in doing so we also identified the molecular mechanisms with which SMG1 is downregulated by its own insertion domain in cis and SMG8 in trans.

## Results

### The compound SMG1i specifically inhibits SMG1 kinase activity in vitro

We set out to study the inhibitory mechanism of SMG1**i** (*Figure 1A*), a small-molecule compound based on a pyrimidine derivative that has been reported to inhibit SMG1 catalytic activity (*Gopalsamy et al., 2012*; *Mino et al., 2019*). We expressed and purified human wild-type SMG1-8-9 complex from piggyBac transposase generated HEK293T stable cell pools, essentially as described before (*Gat et al., 2019*). We have previously shown that the use of a UPF1 peptide comprising the UPF1 phosphorylation site 1078 (UPF1-LSQ) allows monitoring of its specific phosphorylation by SMG1

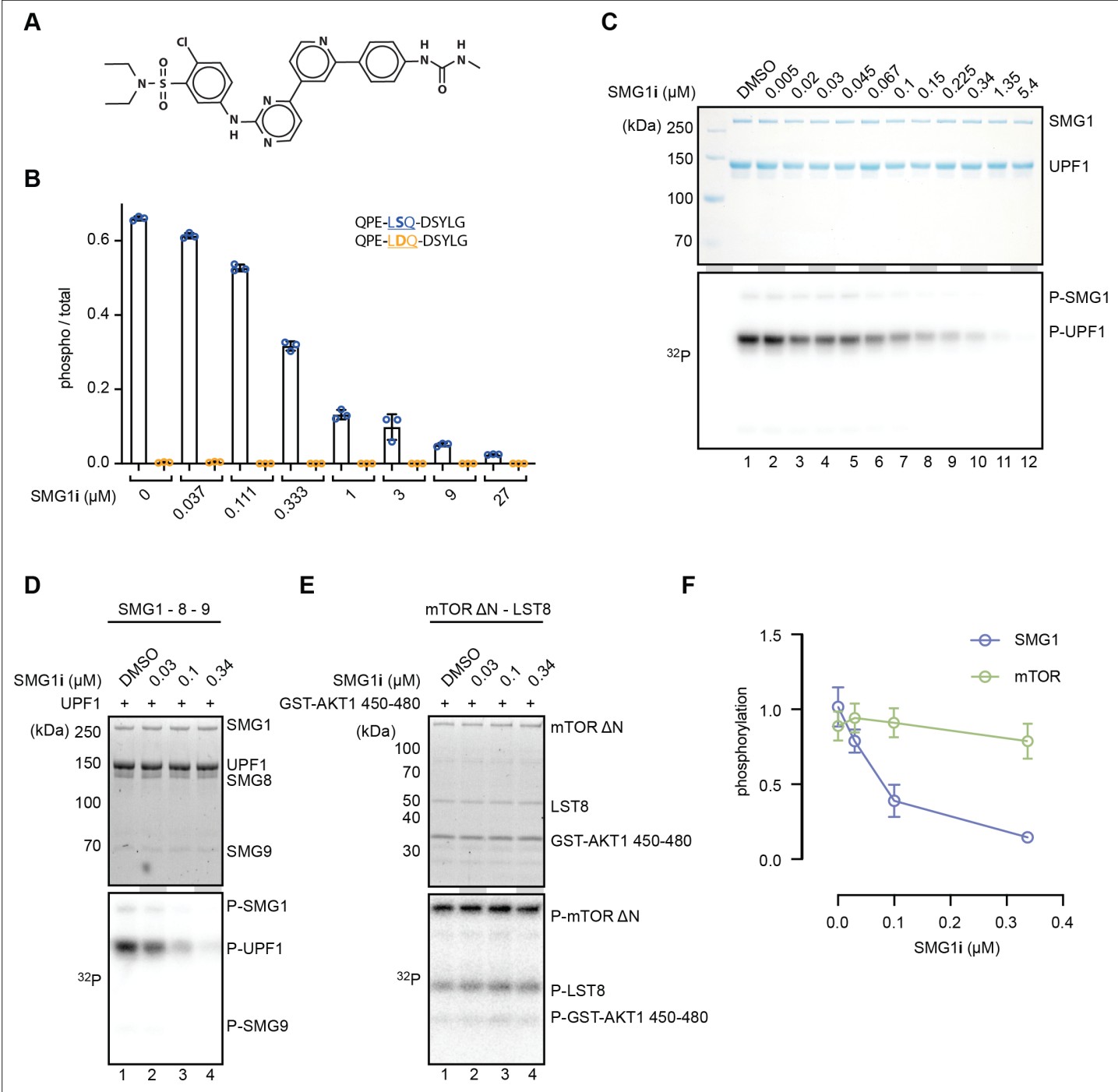

**Figure 1.** SMG1i specifically inhibits SMG1 kinase activity in vitro. (**A**) Structure of the SMG1 inhibitor (SMG1i). (**B**) Titration of SMG1i using a mass spectrometry-based phosphorylation assay with 500 nM SMG1-8-9 and the indicated UPF1-derived peptides as substrates. (**C**) Titration of SMG1i using a radioactivity-based phosphorylation assay with 100 nM SMG1-8-9 and full-length UPF1 as a substrate. The Coomassie-stained gel is shown on top, and the corresponding radioactive signal is below. (**D**, **E**) Titration of SMG1i using a radioactivity-based phosphorylation assay with either SMG1-8-9 and full-length UPF1 (**D**) or mTOR^ΔN-LST8 and GST-AKT1 (**E**) as a substrate. Stain-free gels are shown on top and the corresponding radioactive signal at the bottom (*Ladner et al., 2004*). (**F**) Quantification of normalized UPF1 or mTOR (auto-)phosphorylation in the presence of increasing amounts of SMG1i for both SMG1 and mTOR. For each data point, the mean is shown with standard deviations of the three replicates indicated.

The online version of this article includes the following figure supplement(s) for figure 1:

**Source data 1.** Unedited images for gels shown in *Figure 1*.

**Figure supplement 1.** Characterization of SMG1 inhibitor.

*Figure 1 continued on next page*

*Figure 1 continued*

**Figure supplement 2.** Radioactivity-based phosphorylation assays using SMG1 inhibitor with SMG1 and mTOR.

**Figure supplement 2—source data 1.** Unedited images for gels shown in *Figure 1—figure supplement 2*.

over time using mass spectrometry (*Langer et al., 2020*). We made use of this method to assay the effect and potency of SMG1**i** (*Figure 1*, *Figure 1—figure supplement 1*). To avoid artifacts caused by ATP depletion, we performed an ATP titration and analyzed the end-point measurements (*Figure 1—figure supplement 1B*). As a control, we used a UPF1-LDQ peptide in which we exchanged the phosphor-acceptor Ser1078 of UPF1-LSQ to aspartic acid. Next, we repeated end-point measurements under conditions of stable ATP concentration and increasing amounts of SMG1**i** (*Figure 1B*). In this assay, we observed a significant reduction of UPF1-LSQ phosphorylation when adding SMG1**i** at concentrations equimolar to the enzyme (*Figure 1B* and *Figure 1—figure supplement 1C*). To corroborate these results, we repeated titration of SMG1**i** under conditions of stable ATP concentration using a radioactivity-based phosphorylation assay and full-length UPF1 as a substrate. Again, we observed a reduction in UPF1 phosphorylation in the presence of low micromolar amounts of SMG1**i** (*Figure 1C*).

To assess the specificity of SMG1**i**, we tested its effect on the mTOR kinase using a similar radio-active kinase assay. In this experiment, we used a truncated mTOR$^{\Delta N}$-LST8 complex previously shown to be constitutively active and responsive to mTOR inhibitors (*Yang et al., 2013*). As an mTOR kinase substrate, we used a GST-AKT1$^{450–480}$ fusion protein. We observed that SMG1**i** only affected mTOR activity in the highest concentration range, well beyond concentrations that had an observable effect

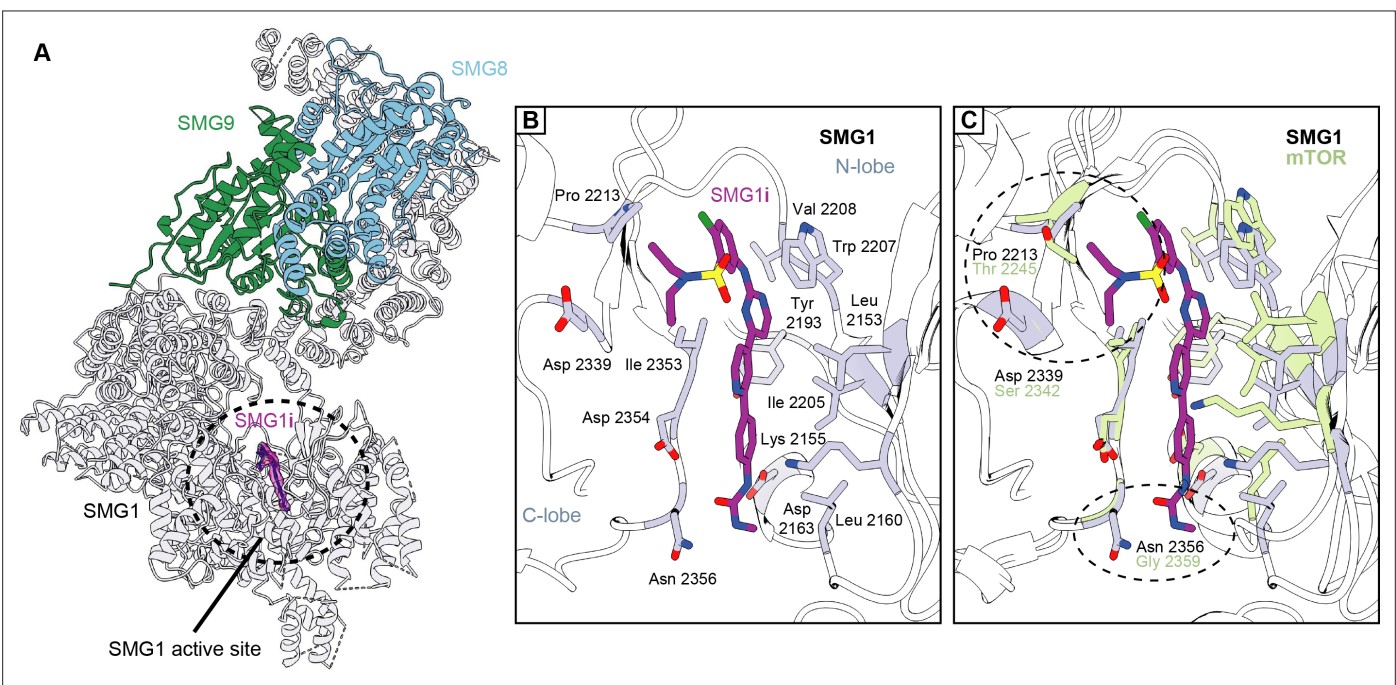

**Figure 2.** Structural basis for selective targeting of SMG1 by the SMG1 inhibitor. (**A**) Model of the SMG1-8-9 kinase complex bound to SMG1**i**. SMG1 is in gray, SMG8 is in blue, and SMG9 is shown in green. SMG1**i** is shown as a magenta model overlaid with the isolated transparent density. Approximate location of the SMG1 active site is indicated by a black circle. (**B**) Key interactions of SMG1**i** with SMG1 active site residues. Important residues located in either the N- or the C-lobe of the SMG1 kinase domain are colored gray. Other parts of SMG1 are transparent and interactions of SMG1**i** with SMG1 backbone are not shown. (**C**) Superposition of SMG1**i**-bound SMG1 with the mTOR active site (PDB identifier: 4JSP) over the catalytic loops of both kinases. Key SMG1 residues indicated in (**B**) are shown alongside the respective mTOR residues colored in green. Regions possibly accounting for preferential interaction of SMG1**i** with SMG1 over mTOR are circled and the relevant residues are labeled.

The online version of this article includes the following figure supplement(s) for figure 2:

**Figure supplement 1.** Resolution distribution and isotropy of SMG1-centered cryo-EM maps bound to SMG1**i**.

**Figure supplement 2.** Cryo-EM data processing of SMG1**i** data set.

**Figure supplement 3.** Further details of SMG1**i** binding and specificity.

on SMG1 activity (*Figure 1—figure supplement 2A*). We selected four different concentrations of SMG1**i** based on these assays (*Figure 1C*, *Figure 1—figure supplement 2A*) and repeated the radioactivity-based kinase assay for both SMG1 and mTOR in triplicates (*Figure 1D and E*, *Figure 1— figure supplement 2B, C*). Upon quantifying the levels of phosphorylation using densitometry and normalizing them to the amount of protein loaded in each lane, we found that SMG1**i** robustly inhib-ited SMG1 while mTOR activity was only weakly affected (*Figure 1F*, *Figure 1—figure supplement 2D, E,F*). In agreement with previous results (*Gopalsamy et al., 2012*), we concluded that SMG1**i** displays high potency and specificity in inhibiting the SMG1 kinase in vitro.

## Cryo-EM structure of SMG1-8-9 bound to the SMG1i inhibitor

To understand the molecular basis for the SMG1**i** mode of action, we reconstituted purified wild-type SMG1-8-9 with an excess of SMG1**i** and subjected the sample to single-particle cryo-EM analysis. The resulting reconstruction reached an overall resolution of 3.1 Å, and we observed clear density for the SMG1 inhibitor bound to the kinase active site (*Figure 2A*, *Figure 2—figure supplements 1 and 2* and *Figure 2—figure supplement 3B, C*). Superposition with our previously published AMPPNP- and substrate-bound model (*Langer et al., 2020*) revealed that SMG1**i** exploits essentially the same binding site as the ATP analog. However, SMG1**i** wedges deeper in between the N- and C-lobe of the kinase domain and is engaged in more extensive interactions (*Figure 2B* and *Figure 2—figure supplement 3A and D*). These observations identify SMG1**i** as a potent ATP-competitive inhibitor.

While many of the SMG1 residues interacting with SMG1**i** are conserved within the PIKK family, two contact sites appear to be specific: SMG1 Pro2213 and Asp2339 contact the sulfonamide moiety of the inhibitor while Asn2356 forms hydrogen bonds with the phenyl urea/carbamide group (*Figure 2B*). Superposition of the SMG1 active site in the SMG1**i**-bound structure with the active site of mTOR$^{\Delta N}$ shows that the corresponding residues have diverged and would not be able to mediate analogous interactions with the SMG1**i** sulfonamide group (mTOR Thr2245 and Ser2342 at the posi-tions of SMG1 Pro2213 and Asp2339) or with the urea group (mTOR Gly2359 at the position of SMG1 Asn2356) (*Figure 2C*). The same regions of the small molecule are diverging or missing in the mTOR inhibitor Torin 2 (*Figure 2—figure supplement 3E*). Taken together, these observations rationalize the specificity of SMG1**i** for SMG1 over mTOR$^{\Delta N}$ that we observed in the in vitro assays (*Figure 1D-F* and *Figure 1—figure supplement 2D, E and F*).

Superposition with other PIKKs suggests the presence of potentially similar discriminatory inter-actions: like mTOR, DNA-PK lacks a favorable interaction site for the SMG1**i** urea group (Gly3943 at the position of SMG1 Asn2356) (*Figure 2—figure supplement 3F*). In addition, DNA-PK PRD residue Lys4019 would sterically clash with the SMG1**i** urea group in the inactive conformation of the DNA-PK active site. In the case of ATR, the SMG1**i** sulfonamide site has diverged (Gly2385 at the position of SMG1 Pro2213) and residues in the N-lobe would result in a steric clash with the urea group of the inhibitor (Lys2329 and Asp2330 corresponding to SMG1 Leu2157 and Glu2158) (*Figure 2—figure supplement 3G*). Finally, mTOR, DNA-PK, and ATR active sites have a different relative orientation of the kinase N- and C-lobes as compared to SMG1, changing the overall chemical environment at the SMG1**i**-binding site (*Figure 2C* and *Figure 2—figure supplement 3F and G*). We conclude that subtle differences in the SMG1 structure underpin the specificity of the SMG1**i** inhibitor.

## The SMG1 insertion domain contains a PRD that blocks substrate binding in the presence of SMG8

In the reconstruction of the SMG1**i**-bound SMG1-8-9 complex, we observed additional density near the end of the kinase domain, protruding from the position where the insertion domain is expected to start (*Figure 3*, *Figure 3—figure supplement 1A and B*). While the lower resolution for this part of the map was not sufficient to unambiguously build an atomic model, this density clearly docks at the substrate-binding site of the kinase domain (*Figure 3A*, *Figure 3—figure supplement 1A, B and C*). Superposition with the previous cryo-EM structure of SMG1-8-9 complex bound to a UPF1 peptide (*Langer et al., 2020*) revealed that the additional density in the current reconstruction is mutually exclusive with the position of a bound substrate (*Figure 3I*, *Figure 3—figure supplement 1A*). This observation is in good agreement with the reported positions of the PRDs in structures of the auto-inhibited forms of yeast Tel1$^{ATM}$ and human DNA-PK (*Jansma et al., 2020*; *Yates et al., 2020*; *Chen et al., 2021*). We concluded that the SMG1 insertion domain contains a PRD that can directly occupy

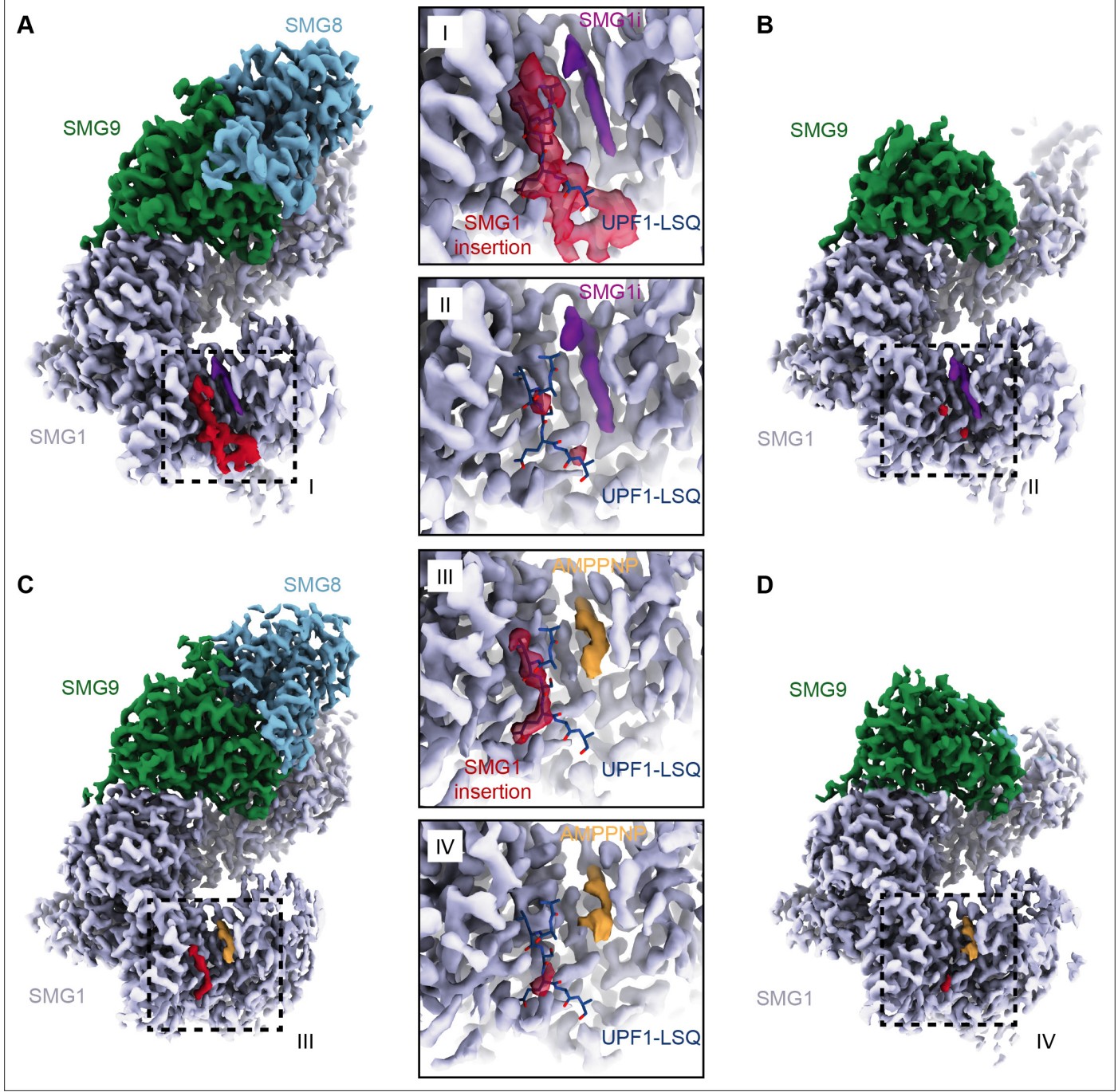

**Figure 3.** Structures of SMG1-8-9 and SMG1-9 complexes reveal that the SMG1 insertion domain can block the substrate-binding path in the presence of SMG8. (**A**) Cryo-EM density of SMG1-8-9 bound to SMG1i. Density for the inhibitor is in magenta, the N-terminus of the SMG1 insertion is in red, and all other parts as indicated. (**B**) Cryo-EM density of SMG1-9 complex bound to the SMG1 inhibitor. Everything else as in (**A**). (**C**) Cryo-EM density of SMG1-8-9 bound to AMPPNP. Density for AMPPNP is in orange and extra density attributed to the SMG1 insertion is in red. (**D**) As in (**C**), but for the SMG1-9 complex bound to AMPPNP. Insets I–IV show close-ups of the indicated kinase active site densities superimposed with the model for the UPF1-LSQ substrate shown as blue sticks (PDB identifier: 6Z3R). All maps segmented. cryo-EM, cryo-electron microscopy.

The online version of this article includes the following figure supplement(s) for figure 3:

**Figure supplement 1.** Details of the SMG1 insertion N-terminus.

**Figure supplement 2.** Resolution distribution and isotropy of SMG1-centered cryo-EM maps bound to AMPPNP.

**Figure supplement 3.** Cryo-EM data processing of AMPPNP data set.

**Figure supplement 4.** Details of the SMG1-9 complex.

and block the substrate-binding path in the kinase active site, analogous to other PIKKs. The SMG1 PRD would have to re-arrange upon substrate binding, as observed in the case of the inactive-to-active transition of DNA-PK (*Chen et al., 2021*).

Within the cryo-EM data set collected on an SMG1-8-9 complex incubated with SMG1**i**, we isolated a subset of particles missing any density for SMG8, hence representing an SMG1**i**-bound SMG1-9 complex (*Figure 3B*, *Figure 2—figure supplement 1C*, *Figure 3—figure supplement 4*). The corresponding 3.6 Å reconstruction not only showed a similar mode of interaction of SMG1**i**, but also several features distinct from SMG1 in isolation and the SMG1-8-9 ternary complex. Globally, the α-solenoid of SMG1 rearranges, consistent with the notion of a stepwise compaction of SMG1 upon binding first SMG9 and then SMG8 (*Figure 3—figure supplement 4A and B*; *Melero et al., 2014*; *Zhu et al., 2019*). Locally, the mode of interaction between SMG9 and the SMG1 α-solenoid differs between the two complexes. In the binary SMG1-9 complex, the portion of SMG9 that interacts with the N-terminal HEAT repeat helices of SMG1 changes conformation, exploiting part of a binding site that was occupied by a short N-terminal segment of SMG8 in the ternary complex (*Figure 3—figure supplement 4C, D and E*). Indeed, two hydrophobic residues of SMG9 (Leu456 and Leu457) take the place normally reserved for two hydrophobic residues of SMG8 (Leu351 and Leu352) (*Figure 3—figure supplement 4F and G*). Hence, the conformation of the SMG9 segment observed in the SMG1-9 complex is incompatible with the binding of SMG8 to this part of SMG1. Importantly, the reconstruction of SMG1-9 showed no ordered density at the active site for the PRD (*Figure 3B and II*, *Figure 3—figure supplement 1A*), which is in contrast to the ordered PRD density visualized in the SMG1**i**-bound SMG1-8-9 complex.

Next, we asked whether the autoinhibitory state of the SMG1 insertion domain in the SMG1-8-9 complex is due specifically to the presence of the SMG inhibitor or whether it is a more general feature. To address this question, we reconstituted purified wild-type SMG1-8-9 with an excess of AMPPNP (instead of SMG1**i**) and subjected the sample to a similar single-particle cryo-EM analysis routine. We obtained maps for SMG1-8-9 and SMG1-9 at overall resolutions of 2.8 Å and 3.1 Å, respectively (*Figure 3C and D*, *Figure 3—figure supplements 2 and 3*). Both reconstructions showed clear density for AMPPNP in the SMG1 active site. In the reconstruction of the ternary SMG1-8-9 complex, we observed an additional density at the substrate-binding site, in the same position as that observed when in the presence of SMG1**i** (*Figure 3III, IV*). However, in the context of AMPPNP this additional density is less prominent and does not connect to the SMG1 kinase domain. It thus appears that the SMG1 inhibitor has a stabilizing effect on the autoinhibitory conformation of the SMG1 PRD. In the reconstruction of the binary SMG1-9 complex in the presence of AMPPNP, we observed no ordered density at the substrate-binding site, consistent with the reconstruction obtained in the presence of SMG1**i**. We concluded that the autoinhibitory conformation of the PRD in the insertion domain of SMG1 is connected to the presence of a nucleotide in the ATP-binding site and is stabilized by the presence of SMG8.

## The SMG1 insertion domain can block overall access to the kinase active site and interacts with the SMG8 C-terminus

The findings above suggested a possible role for SMG8 in stabilizing an autoinhibited state of SMG1. Indeed, previous work implicated SMG8 and, in particular, its C-terminal domain in the downregulation of SMG1 activity (*Zhu et al., 2019*; *Arias-Palomo et al., 2011*; *Deniaud et al., 2015*; *Yamashita et al., 2009*). SMG8 has a modular domain organization: the N-terminal G-domain is followed by a helical stalk that protrudes into solvent with well-defined density, but the remaining C-terminal region (amounting to about 45% of the molecule) is flexible and poorly resolved in the published cryo-EM studies (*Gat et al., 2019*; *Zhu et al., 2019*; *Langer et al., 2020*). By further processing of the SMG1**i**-bound SMG1-8-9 data, we could achieve improved density for the C-terminal domain of SMG8, showing a knob-like feature directly connected to the end of the stalk and in turn connecting to a larger globular mass (*Figure 4A*). Concurrently, we observed an additional density extending from the location of the PRD that we attributed to the SMG1 insertion domain (see below). On one side, this density wraps along the catalytic module of SMG1, occludes the access to the kinase active site (*Figure 4A*), and reaches toward the $IP_6$-binding site (*Figure 4A and C*). On the other side, the density projects into solvent and approaches the globular SMG8 C-terminal region. Although the low resolution of this part of the map did not allow model building, the close proximity suggested a physical

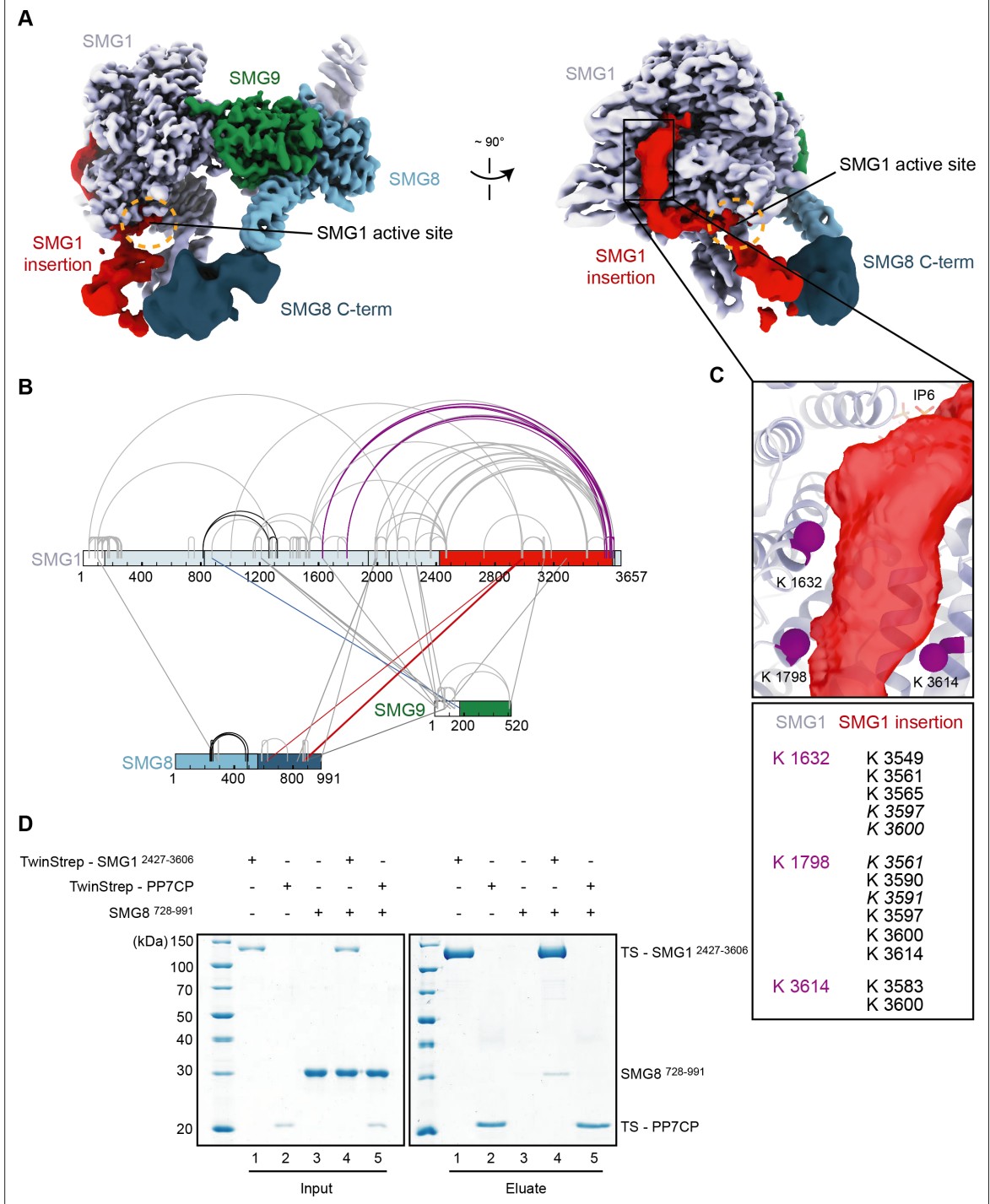

**Figure 4.** The SMG1 insertion domain can block overall access to the kinase active site. (**A**) Cryo-EM map after 3D variability analysis filtered by resolution and segmented. Two different views displaying extra density for SMG8 C-terminus (dark blue) and SMG1 insertion domain. The location of the SMG1 kinase active site is indicated by an orange circle. (**B**) Inter and intra cross-links of the SMG1-8-9 kinase complex. Proteins are colored as in (**A**). Intra-links shown in *Figure 4—figure supplement 1B, C* are in black, the inter-link shown in *Figure 4—figure supplement 1D* is in blue, and inter-links between SMG1 insertion and SMG1 C-terminus are in red. (**C**) Zoom-in highlighting positions of Lys residues (magenta spheres) crosslinking to the C-terminus of the SMG1 insertion domain. A table listing the cross-linked residues is shown below. Cross-links found in only one of two samples are in italics. (**D**) Coomassie-stained SDS-PAGE analysis of pull-down experiment showing an interaction between SMG1 insertion domain (SMG1$^{2427–3606}$) and SMG8 C-terminus (SMG8$^{728–991}$). cryo-EM, cryo-electron microscopy.

The online version of this article includes the following source data and figure supplement(s) for figure 4:

*Figure 4 continued on next page*

*Figure 4 continued*

**Source data 1.** Table with cross-linking data used in *Figure 4*.

**Source data 2.** Unedited images for gels shown in *Figure 4*.

**Figure supplement 1.** Cross-linking mass spectrometry of SMG1-8-9.

**Figure supplement 1—source data 1.** Unedited images for gels shown in *Figure 4—figure supplement 1*.

**Figure supplement 2.** Selected spectra of detected SMG1-8-9 intra-links.

**Figure supplement 3.** Integration of cryo-EM, cross-linking MS, and AlphaFold data reveals a model for the SMG8 C-terminus.

**Figure supplement 4.** Further characterization of SMG1-8-9—centered interactions.

**Figure supplement 4—source data 1.** Unedited images for gels shown in *Figure 4—figure supplement 4*.

contact between the SMG1 insertion domain and the SMG8 C-terminal domain. We proceeded to test the interplay between these two parts of the SMG1-8-9 complex.

We first used cross-linking mass spectrometry (XL-MS) analysis (*Figure 4B*, *Figure 4—figure supplements 1 and 2*). Upon treatment of purified SMG1-8-9 with Bis(sulfosuccinimidyl)suberate (BS³), we detected intra and inter cross-links consistent with parts of the complex for which an atomic model was available and measuring well within the expected range of distance for BS³ cross-links (*Figure 4B*, *Figure 4—figure supplements 1 and 2*). We also observed reproducible intra cross-links between the end of the SMG1 insertion and the catalytic module, in particular the FAT (residues 823–1938) and kinase (residues 2080–2422) domains (*Figure 4B and C*). These cross-links mapped in close proximity to the aforementioned density extending from the PRD. We thus attribute this density to the C-terminus of the SMG1 insertion (*Figure 4A, B and C*). By combining our low-resolution EM density and our cross-linking data with an AlphaFold prediction of SMG8 (*Jumper et al., 2021*), we were able to produce a more complete model of SMG1-8-9, now containing large parts of the SMG8 C-terminus (*Figure 4—figure supplement 3*). Importantly, we detected inter cross-links between the SMG1 insertion domain and the SMG8 C-terminal domain (*Figure 4B*), consistent with the proximity of the EM densities (*Figure 4A*) and previously reported XL-MS data (*Deniaud et al., 2015*).

Next, we tested whether these parts of the complex can interact directly in biochemical assays with recombinant proteins. We purified SMG1 insertion domain (SMG1 residues 2427–3606) with an N-terminal TwinStrep-tag from transiently transfected HEK293T cells and SMG8 C-terminus (SMG8 residues 728–991) from *Escherichia coli*. Since the structural analysis is indicative of a dynamic or weak contact, we performed pull-down assays in low-salt conditions. We observed that the SMG1 insertion domain indeed co-precipitated the SMG8 C-terminus in these conditions (*Figure 4D*). As a control, an unrelated TwinStrep-tagged protein (PP7 coat protein) did not pull down SMG8 C-terminus under the same conditions (*Figure 4D*). As anticipated, increasing the salt concentration had a detrimental effect on the SMG8 C-terminus—SMG1 insertion co-precipitation (*Figure 4—figure supplement 4A*).

Taken together, these experiments indicate the existence not just of proximity but of a direct physical link between the SMG1 insertion domain and SMG8 C-terminus, rationalizing the observed dependency of SMG1 autoinhibition on the presence of SMG8 (*Figure 3*).

## Conclusions

In this manuscript, we reported a reconstruction of the SMG1-8-9 complex bound to an SMG1 inhibitor. We showed that this compound binds to the ATP-binding site within the kinase active site. By comparison with structural data of related kinases, we identified two functional groups within the small molecule that possibly confer specificity to SMG1. Using the SMG1-related mTOR kinase as an example, we confirmed the specific action of the inhibitor biochemically.

In addition to the SMG1 inhibitor, we observed density for the N-terminal part of the SMG1 insertion domain in the SMG1-8-9 reconstruction. This part of the insertion domain acts as a PRD and occupied the substrate-binding path within the SMG1 active site. Consistent with data for other PIKKs, a PRD within the SMG1 insertion domain can therefore restrict access to the kinase active site.

The structure of an SMG1**i**-bound SMG1-9 complex reconstructed from the same data set as SMG1**i**-bound SMG1-8-9 did not show ordered density for the SMG1 PRD in the SMG1 active site. This suggested that an interaction between the SMG1 insertion and SMG8 was important for stabilizing the autoinhibited state of the complex. Consistently, our biochemical analysis indicated a direct physical link between parts of the SMG1 insertion domain and the C-terminal domain of SMG8.

While autoinhibition was not observed in previously published apo-structures, our reconstructions of SMG1-9 and SMG1-8-9 bound to AMPPNP showed that blockage of the substrate-binding path was also possible when the SMG1-8-9 complex active site was bound to a nucleotide. Accordingly, the described autoinhibition was not a phenomenon limited to the inhibitor-bound structures. We conclude that at least two layers of regulation of SMG1 kinase activity exist: first, access to the SMG1 active site can be restricted in cis by a PRD within the SMG1 insertion domain. Second, this autoinhibition is stabilized in trans by the SMG8 C-terminus.

How is the autoinhibited complex activated? The position of the SMG1 insertion domain in the inhibitor-bound SMG1-8-9 complex reconstruction raised the possibility that the autoinhibited complex might not be able to interact with its substrate, UPF1 (*Figures 3I, III and 4A*). To test this hypothesis, we incubated TwinStrep-tagged SMG1-8-9 with increasing concentrations of SMG1i before adding UPF1 and performed a pull-down experiment (*Figure 4—figure supplement 4B*). The efficiency of UPF1 co-precipitation was unaffected by the presence of SMG1i, suggesting that either UPF1 can overcome blockage of the SMG1 active site by the SMG1 insertion domain or has secondary binding sites independent of the observed SMG1 autoinhibition. Indeed, superposition of the autoinhibited complex reconstruction with a previously reported negative-stain map of a cross-linked UPF1-bound SMG1-8-9 complex raised the possibility that UPF1-binding and SMG1 autoinhibition might be able to occur in parallel (*Figure 4—figure supplement 4C*). An alternative explanation would be that optimal positioning of UPF1 with respect to the SMG1 kinase will allow the unstructured N- and C-terminal ends of UPF1 to efficiently compete with the SMG1 insertion domain for binding to the SMG1 active site in vivo. This would be mimicked by the high protein concentrations used in in vitro assays (such as *Figure 4—figure supplement 4B*).

Whether SMG8 or an SMG8-9 complex dissociates from SMG1 at any point in the kinases' catalytic cycle during NMD in vivo is an outstanding question. Early biochemical experiments have suggested that SMG8 plays a crucial role in recruiting SMG1 complex to the NMD-competent messenger ribonucleoprotein particle (*Yamashita et al., 2009*). Given that the SMG1-8-9 complex actively phosphorylated UPF1 in in vitro experiments using purified proteins and our observation that SMG1i binding did not impair the interaction of SMG1 with UPF1, it is tempting to speculate that UPF1 itself might be sufficient to overcome the effect of SMG8 on stabilizing SMG1 autoinhibition within an SMG1-8-9 complex. In this model, nucleotide binding and kinase activation could occur in two distinct steps during NMD: upon ATP binding, the kinase complex would adopt an autoinhibited conformation and

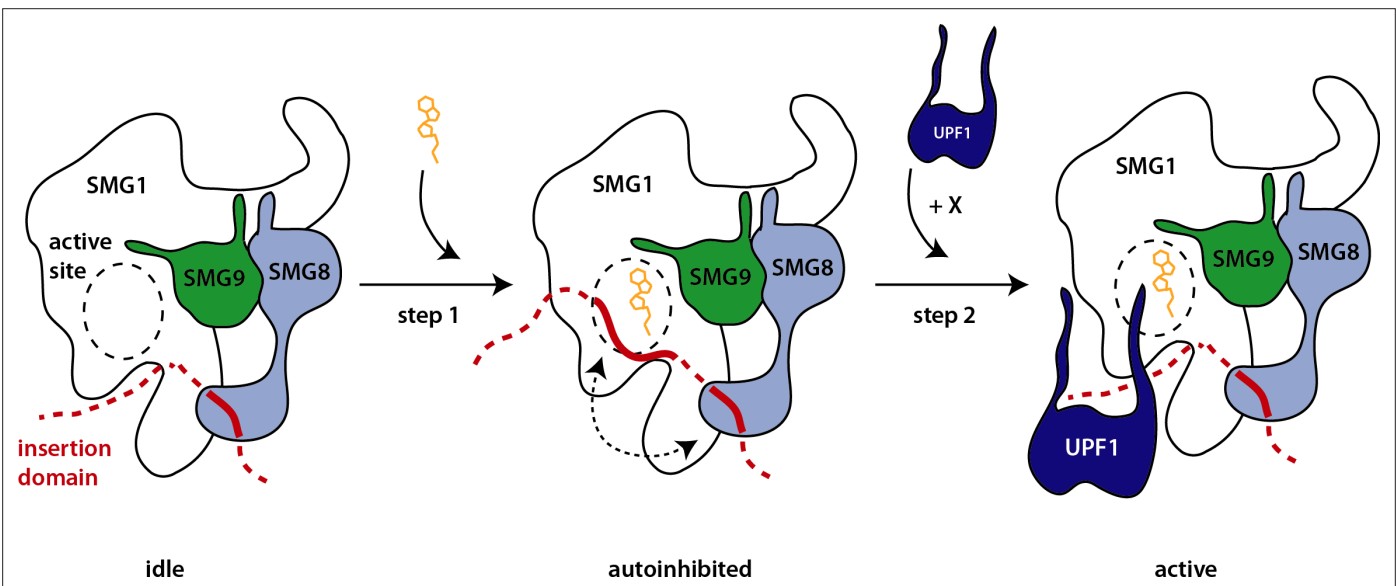

**Figure 5.** Hypothetical model of SMG1 kinase regulation. Structural and biochemical data suggest different layers of regulation on SMG1 kinase activity. Upon ATP binding (orange), the SMG1 kinase adopts an autoinhibited conformation (step 1)—mediated by the concerted action of the SMG1 insertion domain in cis and the SMG8 C-terminus in trans (dotted arrow). The presence of the correct substrate and likely other factors/cues (indicated by X) then trigger the release of the autoinhibition and activity of the kinase toward UPF1 (step 2).

only be licensed for specific phosphorylation of its target by the substrate itself (*Figure 5*). In addition, autophosphorylation of the kinase complex may play a crucial role in kinase activation.

Taken together, our structures will help to improve the design of selective PIKK active site inhibitors, for example, specific for SMG1. Our data suggest a unifying structural model for concerted action of the SMG1 insertion domain in cis and SMG8 in trans to tune SMG1 kinase activity, shedding light on the intricate regulation of this kinase complex in metazoan NMD.

# Materials and methods

## Key resources table

| Reagent type (species) or resource | Designation | Source or reference | Identifiers | Additional information |
|---|---|---|---|---|
| Gene (*Homo sapiens*) | SMG1 | Shigeo Ohno lab | Uniprot Q96Q15 | |
| Gene (*H. sapiens*) | SMG8 | Shigeo Ohno lab | Uniprot Q8ND04 | |
| Gene (*H. sapiens*) | SMG9 | Shigeo Ohno lab | Uniprot Q9H0W8 | |
| Cell line (*H. sapiens*) | HEK293T | ATCC | | |
| Strain, strain background (*Escherichia coli*) | BL21 Star (DE3) pRARE | EMBL Heidelberg Core Facility | | Electrocompetent cells |
| Peptide, recombinant protein | UPF1-LSQ (peptide 1078) and derivative | In-house as described in doi: https://elifesciences.org/articles/57127 | | |
| Chemical compound, drug | SMG1 inhibitor | Robert Bridges, Rosalind Franklin University of Medicine and Science, and the Cystic Fibrosis Foundation | | |
| Chemical compound, drug | AMPPNP | Sigma-Aldrich | | |
| Chemical compound, drug | ATP | Sigma-Aldrich | | |
| Software, algorithm | SerialEM | https://bio3d.colorado.edu/SerialEM/ | | |
| Software, algorithm | Focus | https://focus.c-cina.unibas.ch/ wiki/doku.php | v1.1.0 | |
| Software, algorithm | RELION | doi: 10.7554/eLife.42166 | RELION 3.0 | |
| Software, algorithm | Cryosparc | doi: 10.1038/nmeth.4169 | Cryosparc2 | |
| Software, algorithm | UCSF Chimera | UCSF, https://www.cgl.ucsf.edu/chimera/ | | |
| Software, algorithm | UCSF ChimeraX | UCSF, https://www.rbvi.ucsf.edu/chimerax/ | | |
| Software, algorithm | COOT | http://www2.mrc-lmb.cam.ac.uk/personal/pemsley/coot/ | | |
| Software, algorithm | PHENIX | https://www.phenix-online.org/ | PHENIX 1.17 | |
| Software, algorithm | Molprobity | Duke Biochemistry, http://molprobity.biochem.duke.edu/ | | |
| Software, algorithm | PyMOL | PyMOL Molecular Graphics System, Schrodinger LLC | PyMOL 2.3.2 | |

## Protein expression and purification

TwinStrep-tagged SMG1-8-9 complex was expressed and purified as described before (*Gat et al., 2019*; *Langer et al., 2020*). The source cells were authenticated by genotyping (Eurofins) and tested negative for mycoplasma contamination (LookOut Mycoplasma PCR Detection Kit).

N-terminally TwinStrep-tagged SMG1 insertion domain (SMG1$^{2427–3606}$) was expressed and purified essentially identical to the SMG1-8-9 full-length complex, with the exception that lysis and affinity purification was carried out in a buffer based on 2× phosphate-buffered saline (PBS). The protein was frozen in gel filtration buffer at concentrations ranging from 15 to 40 μM depending on the purification.

Untagged full-length UPF1 expressed in *E. coli* was purified as reported previously (*Langer et al., 2020*).

N-terminally TwinStrep-tagged full-length UPF1 was expressed using a stable pool of HEK293T cells as described for SMG1-8-9. Approximately $400 \times 10^6$ cells were lysed with a Dounce homogenizer in buffer containing 1× PBS, 5 mM $MgCl_2$, 1 µM $ZnCl_2$, 10% (v/v) glycerol, 1 mM DTT supplemented for lysis with Benzonase and DNase I as well as EDTA-free Complete Protease Inhibitor Cocktail (Roche). The lysate was cleared by centrifugation at 25,000 rpm for 30 min at 10°C. The supernatant was filtered and applied to a StrepTrap HP column (Sigma-Aldrich). After passing 25 column volumes of lysis buffer without supplements, the column was further washed with 25 column volumes of buffer Hep A (20 mM HEPES/NaOH pH 7.4, 85 mM KCl, 2 mM $MgCl_2$, 1 µM $ZnCl_2$, 10% (v/v) glycerol, and 1 mM DTT) and bound protein was eluted directly onto a HiTrap Heparin column (GE Healthcare) using buffer Hep A supplemented with 2.5 mM Desthiobiotin. After washing the Heparin column with 10 column volumes of buffer Hep A, bound protein was eluted using a gradient increasing the salt concentration to 500 mM KCl over 100 ml while collecting fractions (AEKTA prime FPLC system, GE Healthcare). Fractions were analyzed by SDS-PAGE and those containing pure full-length Twin-Strep-UPF1 were collected, concentrated to 30–80 µM, and flash-frozen without buffer exchange using liquid nitrogen until further usage.

The SMG8 C-terminus (SMG8[728–991]) was N-terminally fused to a 6xHis-SUMO tag and expressed in *E. coli* BL21 STAR (DE3) pRARE cells. Bacteria were grown in TB medium at 37°C shaking at 180 rpm. At an OD of 2, cultures were cooled down to 18°C and overnight protein expression was induced by adding IPTG. After harvesting (6000 rpm, 10 min), bacteria were lysed by sonication in a buffer containing 20 mM Tris-HCl pH 7.5, 400 mM NaCl, 10 mM Imidazole pH 7.5, 1 mM ß-mercaptoethanol, Benzonase, DNase I and 1 mM PMSF. After centrifugation (25,000 rpm, 30 min, 10°C) and filtration, the cleared lysate was applied to a HisTrap FF column (Cytiva) and the column was washed with 30 column volumes of wash buffer (20 mM Tris-HCl pH 7.5, 400 mM NaCl, 40 mM Imidazole pH 7.5, and 1 mM ß-mercaptoethanol). Bound protein was eluted with wash buffer supplemented with 340 mM Imidazole pH 7.5 and the elution was combined with His-tagged SUMO protease for cleavage of the N-terminal tag and dialyzed overnight against 20 mM Tris-HCl pH 7.5, 400 mM NaCl, and 1 mM ß-mercaptoethanol. The dialyzed, protease treated sample was passed again over a 5 ml HisTrap FF column to remove cleaved tags and protease, the flow-through was diluted to 150 mM NaCl and 10°C (v/v) glycerol and applied to a HiTrap SP HP column (Cytiva). After washing with 10 column volumes of 20 mM Tris-Cl pH 7.5, 150 mM NaCl, 10°C (v/v) glycerol, and 1 mM ß-mercaptoethanol, proteins were eluted using a gradient increasing the salt concentration to 500 mM NaCl. The peak fractions containing SMG8[728–991] were pooled, concentrated, and injected onto a Superdex 75 10/300 GL (Cytiva) equilibrated with 20 mM HEPES/NaOH pH 7.4, 200 mM NaCl, and 1 mM DTT. Again, the peak fraction containing SMG8[728–991] was pooled and concentrated to 100–200 µM and flash-frozen in gel filtration buffer using liquid nitrogen. mTOR[ΔN]-LST8 and GST-AKT1[450–480] were prepared as described before (*Gat et al., 2019*).

## Kinase assays

The mass spectrometry-based kinase assays were carried out as previously reported (*Langer et al., 2020*). For experiments involving SMG1**i** or Torin 2, the assembled reactions were incubated in the absence of ATP for 30 min at 4°C. All reactions involving SMG1**i** were performed with equal amounts of DMSO present.

Kinase assay based on the use of $\gamma$-$^{32}$P-labeled ATP was carried out as reported. For all assays, 100 nM of kinase complex and 1 µM of substrate were used. All reactions were supplemented with equal amounts of DMSO and incubated without ATP for 30 min at 4°C before starting the experiment. After 30 min at 30°C, phosphorylation reactions were stopped by adding SDS-containing sample buffer and samples were analyzed using an SDS-polyacrylamide gel supplemented with 2,2,2-trichlorethanol (TCE, Acros Organics) for stain-free visualization (*Ladner et al., 2004*). The TCE-tryptophan reaction was started by exposing the protein gel to UV light for 2 min and proteins were visualized and quantified using a Bio-Rad gel imaging system and the Image Lab software (version 6.1, Bio-Rad) or by Coomassie-staining. Subsequently the gel was washed several times with water to remove residual $\gamma$-$^{32}$P-labeled ATP and phosphoproteins were detected using autoradiography and an Amersham Typhoon RGB biomolecular imager (GE Healthcare). Phosphoprotein signal (P-UPF1 or P-mTOR[ΔN]) was quantified by densitometry using Fiji and normalized to the total amount of protein

determined using TCE. Independent triplicates of each condition were normalized to the DMSO-only sample and plotted using Prism (GraphPad).

## Cross-linking mass spectrometry

For cross-linking mass spectrometry, 1 µM of SMG1-8-9 complex was incubated with 0.5 mM $BS^3$ for 30 min on ice in a buffer containing 1× PBS, 5 mM $MgCl_2$, and 1 mM DTT. The reaction was quenched by adding 40 mM Tris-HCl pH 7.9 and incubating for 20 min on ice. The sample was spun for 15 min at 18,000×$g$. For denaturation of the crosslinked proteins, 4 M Urea and 50 mM Tris were added to the supernatant and the samples were sonicated using a Bioruptor Plus sonication system (Diogenode) for 10×30 s at high intensity. For reduction and alkylation of the proteins, 40 mM 2-cloroacetamide (CAA, Sigma-Aldrich) and 10 mM Tris (2-carboxyethyl)phosphine (TCEP; Thermo Fisher Scientific) were added. After incubation for 20 min at 37°C, the samples were diluted 1:2 with MS grade water (VWR). Proteins were digested overnight at 37°C by addition of 1 µg of trypsin (Promega). Thereafter, the solution was acidified with trifluoroacetic acid (TFA; Merck) to a final concentration of 1%, followed by desalting of the peptides using Sep-Pak C18 1 cc vacuum cartriges (Waters). The elution was vacuum dried.

Enriched peptides were loaded onto a 30 cm analytical column (inner diameter: 75 µm; packed in-house with ReproSil-Pur C18-AQ 1.9 µm beads, Dr. Maisch GmbH) by the Thermo Easy-nLC 1000 (Thermo Fisher Scientific) with buffer A ( 0.1% (v/v) Formic acid) at 400 nl/min. The analytical column was heated to 60°C. Using the nanoelectrospray interface, eluting peptides were sprayed into the benchtop Orbitrap Q Exactive HF (Thermo Fisher Scientific)(*Hosp et al., 2015*). As gradient, the following steps were programmed with increasing addition of buffer B ( 80% Acetonitrile, 0.1% Formic acid): linear increase from 8% to 30% over 60 min, followed by a linear increase to 60% over 5 min, a linear increase to 95% over the next 5 min, and finally maintenance at 95% for another 5 min. The mass spectrometer was operated in data-dependent mode with survey scans from $m/z$ 300to 1650 Th (resolution of 60k at $m/z$=200 Th), and up to 15 of the most abundant precursors were selected and fragmented using stepped Higher-energy C-trap Dissociation (HCD with a normalized collision energy of value of 19, 27, and 35). The MS2 spectra were recorded with dynamic $m/z$ range (resolution of 30k at $m/z$=200 Th). AGC target for MS1 and MS2 scans were set to 3×$10^6$ and $10^5$, respectively, within a maximum injection time of 100 and 60 ms for the MS1 and MS2 scans, respectively. Charge state 2 was excluded from fragmentation to enrich the fragmentation scans for cross-linked peptide precursors.

The acquired raw data were processed using Proteome Discoverer (version 2.5.0.400) with the XlinkX/PD nodes integrated (*Klykov et al., 2018*). To identify the crosslinked peptide pairs, a database search was performed against a FASTA containing the sequences of the proteins under investigation. DSS was set as a crosslinker. Cysteine carbamidomethylation was set as fixed modification and methionine oxidation and protein N-term acetylation were set as dynamic modifications. Trypsin/P was specified as protease and up to two missed cleavages were allowed. Furthermore, identifications were only accepted with a minimal score of 40 and a minimal delta score of 4. Otherwise, standard settings were applied. Filtering at 1 % false discovery rate (FDR) at peptide level was applied through the XlinkX Validator node with setting simple. A file listing identified cross-links can be found as *Figure 4—source data 1*.

## Pull-down assays

Pull downs including the SMG8 C-terminus were performed using Strep-TactinXT Superflow beads (IBA) equilibrated with 20 mM HEPES/NaOH pH 7.4, 50 mM NaCl, 10% (w/v) glycerol, 1 mM DTT, and 0.1% (w/v) NP-40 substitute (Fluka) unless stated otherwise. 1.5 µM TwinStrep-tagged protein was combined with 15 µM untagged protein in the buffer described above and incubated at 4°C for 30 min before adding equilibrated resin. The complete reaction was incubated for 1.5 hr, beads were washed four times with 20× resin volume of buffer, and bound protein eluted by adding buffer supplemented with 50 mM Biotin.

To analyze the effect of SMG1i binding to SMG1 on the interaction between the SMG1-8-9 kinase complex and UPF1, 0.12 µM TwinStrep-tagged SMG1-8-9 was combined with DMSO, or the respective amounts of SMG1i in buffer containing 1× PBS, 5 mM $MgCl_2$, 1 mM DTT, and 0.1% (v/v) NP-40. After incubation for 30 min at 4°C, 0.4 µM of untagged UPF1 was added and the reactions were combined

with MagStrep 'type3' XT beads (IBA). Samples were incubated for 30 min before washing four times with 20× resin volume of buffer and precipitation of bound proteins using SDS-containing sample buffer.

For all pull-down assays, input and elution samples were analyzed by SDS-PAGE and stained with Coomassie.

### Cryo-EM sample preparation, data collection, and data processing

Grids were prepared as described before (*Langer et al., 2020*), with the difference that SMG1-8-9 was incubated with either 4 µM SMG1i or 1 mM AMPPNP for 30 min on ice in 1× PBS, 5 mM MgCl$_2$, and 1 mM DTT before adding 0.04% (v/v) n-octyl-ß-D-glucoside and plunging using an ethane/propane mixture and a Thermo Fisher FEI Vitrobot IV.

Cryo-EM data were essentially acquired as reported previously using a Thermo Fisher FEI Titan Krios G3 microscope equipped with a post-column GIF (energy width 20 eV). The Gatan K3 camera was used in counting mode and data were acquired using SerialEM (*Mastronarde, 2005*) and a beam-tilt based acquisition scheme. The nominal magnification during data collection for both data sets was 105,000× , corresponding to a pixel size of 0.8512 Å at the specimen level. The SMG1i sample was imaged with a total exposure of 89.32 e$^-$/Å$^2$ evenly spread over 4 s and 40 frames. The AMPPNP sample was imaged with a total exposure of 60.99 e$^-$/Å$^2$ evenly spread over 5.7 s and 40 frames at CDS mode. For both data collections, the target defocus ranged between –0.5 and –2.9 µm.

Movies were pre-processed on the fly using Focus (*Biyani et al., 2017*), while automatically discarding images of poor quality. Picked candidate particles were extracted in RELION 3.1 (*Zivanov et al., 2018*; *Scheres, 2012*). After two rounds of reference-free 2D classification, particles were imported to CryoSPARC v2 (*Punjani et al., 2017*) for further processing. After additional sorting in 2D, particles were iteratively classified in 3D to separate SMG1-8-9 and SMG1-9. Particles were distributed over multiple batches for all classification steps. To visualize highly flexible parts of the SMG1 insertion domain in the SMG1i-bound complex, the particle stack containing SMG1-8-9 data was submitted to 3D variability analysis into six clusters. Filtering according to local resolution was employed to better visualize parts of the complex with different resolutions after refinement. Following Ctf refinement, a mask was generated based on the more rigid SMG1 body to focus refinement of the cleaned SMG1i-bound SMG1-8-9 particles on the inhibitor-bound SMG1 active site. For details refer to *Figure 2—figure supplement 2* and *Figure 3—figure supplement 3*. 3D FSC curves were calculated using the 3D FSC online application (*Tan et al., 2017*). Map versus model FSCs were calculated within the PHENIX software suit (*Adams et al., 2011*; *Liebschner et al., 2019*).

### Model building

Model building was carried out using Coot (version 0.8.9.2) (*Emsley et al., 2010*; *Emsley and Cowtan, 2004*) and iterative rounds of real-space refinement in the PHENIX software suit (*Adams et al., 2011*; *Liebschner et al., 2019*) and was based on our previously published models of SMG1-8-9 (PDB identifiers: 6SYT, 6Z3R). Geometric restraints for the SMG1i molecule were calculated with eLBOW (*Moriarty et al., 2009*). For details see *Supplementary file 1*. Structure visualization and analysis were carried out using UCSF ChimeraX (version 1.2.5) (*Goddard et al., 2018*) and PyMOL (version 2.3.2).

To include the AlphaFold derived SMG8 C-terminus model, we downloaded the predicted model for full-length SMG8 from the AlphaFold Protein Structure Database (*Jumper et al., 2021*). We removed all parts with a per-residue confidence score below 50 and superimposed the prediction with our EM data-derived model for the SMG8 N-terminal half (residues 4–559) using Coot. After fusing the roughly positioned AlphaFold-predicted SMG8 C-terminal half (560–991) to our SMG1i-bound SMG-1-8-9 model (PDB identifier: 7PW4), we rigid body-fitted the SMG8 C-terminus into the EM density (resulting in PDB 7PW5). It is worth noting that the somewhat different orientation of the SMG8 C-terminal domain with respect to the N-terminal part of the molecule observed when comparing the EM density to the AlphaFold model before rigid body-fitting is in agreement with the flexibility of this domain generally present in the cryo-EM data. An overview of refinement statistics is shown in *Supplementary file 1*.

### Acknowledgements

The authors are grateful to Robert Bridges, Rosalind Franklin University of Medicine and Science, and the Cystic Fibrosis Foundation for providing the SMG1 inhibitor. Daniel Bollschweiler and Tillman

Schäfer at the MPIB cryo-EM facility for help with EM data collection and Barbara Steigenberger and Elisabeth Weyher at MPIB biochemistry core facility for carrying out mass spectrometry. The authors thank Christian Benda and J Rajan Prabu for maintenance and development of computational infrastructure for EM data processing, Marcela Cueto for TwinStrep-PP7CP, Daniela Wartini for help with tissue culture, and Courtney Long and members of the lab for help with the preparation of the manuscript. This work was supported by funding from the Max Planck Gesellschaft, the European Commission (ERC Advanced Investigator Grant EXORICO), the Novo Nordisk Foundation (Exo-Adapt Center) and the German Research Foundation (DFG SFB1035, GRK1721, SFB/TRR 237) to EC and a Boehringer Ingelheim Fonds PhD fellowship to LL.

## Additional information

### Funding

| Funder | Grant reference number | Author |
|---|---|---|
| Boehringer Ingelheim Fonds | | Lukas M Langer |
| Max-Planck-Gesellschaft | | Elena Conti |
| European Commission | ERC Advanced Investigator Grant EXORICO | Elena Conti |
| Deutsche Forschungsgemeinschaft | SFB1035 | Elena Conti |
| Deutsche Forschungsgemeinschaft | GRK1721 | Elena Conti |
| Deutsche Forschungsgemeinschaft | SFB/TRR 237 | Elena Conti |
| Novo Nordisk Foundation | Exo-Adapt Center | Elena Conti |

The funders had no role in study design, data collection and interpretation, or the decision to submit the work for publication.

### Author contributions

Lukas M Langer, Conceptualization, Data curation, Formal analysis, Investigation, Methodology, Validation, Visualization, Writing - original draft, Writing - review and editing; Fabien Bonneau, Data curation, Formal analysis, Methodology, Writing - review and editing; Yair Gat, Investigation, Writing - review and editing; Elena Conti, Conceptualization, Project administration, Resources, Supervision, Writing - original draft, Writing - review and editing

### Author ORCIDs

Lukas M Langer http://orcid.org/0000-0002-9977-2427
Fabien Bonneau http://orcid.org/0000-0001-8787-7662
Yair Gat http://orcid.org/0000-0002-2338-9384
Elena Conti http://orcid.org/0000-0003-1254-5588

### Decision letter and Author response

Decision letter https://doi.org/10.7554/eLife.72353.sa1
Author response https://doi.org/10.7554/eLife.72353.sa2

## Additional files

### Supplementary files

- Supplementary file 1. Cryo-EM data collection, refinement and validation statistics.
- Transparent reporting form

## Data availability

Models have been deposited in the PDB under the accession codes 7PW4 (SMG1-8-9 bound to SMG1i), 7PW5 (SMG1-8-9 bound to SMG1i, with SMG8 C-terminus), 7PW6 (SMG1 body bound to SMG1i), 7PW7 (SMG1-9 bound to SMG1i), 7PW8 (SMG1-8-9 bound to AMPPNP) and 7PW9 (SMG1-9 bound to AMPPNP). The corresponding EM maps have been deposited in the EMDB under the accession codes EMD-13674, EMD-13675, EMD-13676, EMD-13677, EMD-13678 and EMD-13679.

The following dataset was generated:

| Author(s) | Year | Dataset title | Dataset URL | Database and Identifier |
|---|---|---|---|---|
| Langer LM, Conti E | 2021 | Human SMG1-8-9 kinase complex bound to a SMG1 inhibitor | https://www.rcsb.org/structure/7PW4 | RCSB Protein Data Bank, 7PW4 |
| Langer LM, Conti E | 2021 | Human SMG1-8-9 kinase complex bound to a SMG1 inhibitor with predicted SMG8 C-terminus | https://www.rcsb.org/structure/7PW5 | RCSB Protein Data Bank, 7PW5 |
| Langer LM, Conti E | 2021 | Human SMG1-8-9 kinase complex bound to a SMG1 inhibitor - SMG1 body | https://www.rcsb.org/structure/7PW6 | RCSB Protein Data Bank, 7PW6 |
| Langer LM, Conti E | 2021 | Human SMG1-9 kinase complex bound to a SMG1 inhibitor | https://www.rcsb.org/structure/7PW7 | RCSB Protein Data Bank, 7PW7 |
| Langer LM, Conti E | 2021 | Human SMG1-8-9 kinase complex bound to AMPPNP | https://www.rcsb.org/structure/7PW8 | RCSB Protein Data Bank, 7PW8 |
| Langer LM, Conti E | 2021 | Human SMG1-9 kinase complex bound to AMPPNP | https://www.rcsb.org/structure/7PW9 | RCSB Protein Data Bank, 7PW9 |
| Langer LM, Conti E | 2021 | Human SMG1-8-9 kinase complex bound to a SMG1 inhibitor | https://www.ebi.ac.uk/pdbe/entry/emdb/EMD-13674 | Electron Microscopy Data Bank, EMD-13674 |
| Langer LM, Conti E | 2021 | Human SMG1-8-9 kinase complex with AlphaFold predicted SMG8 C-terminus, bound to a SMG1 inhibitor | https://www.ebi.ac.uk/emdb/error/entry/EMD-13675 | Electron Microscopy Data Bank, EMD-13675 |
| Langer LM, Conti E | 2021 | Human SMG1-8-9 kinase complex bound to a SMG1 inhibitor - SMG1 body | https://www.ebi.ac.uk/emdb/error/entry/EMD-13676 | Electron Microscopy Data Bank, EMD-13676 |
| Langer LM, Conti E | 2021 | Human SMG1-9 kinase complex bound to a SMG1 inhibitor | https://www.ebi.ac.uk/emdb/error/entry/EMD-13677 | Electron Microscopy Data Bank, EMD-13677 |
| Langer LM, Conti E | 2021 | Human SMG1-8-9 kinase complex bound to AMPPNP | https://www.ebi.ac.uk/emdb/error/entry/EMD-13678 | Electron Microscopy Data Bank, EMD-13678 |
| Langer LM, Conti E | 2021 | Human SMG1-9 kinase complex bound to AMPPNP | https://www.ebi.ac.uk/emdb/error/entry/EMD-13679 | Electron Microscopy Data Bank, EMD-13679 |

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
