## [Editor Report]

This study uses CryoEM and biochemical studies to uncover a new and potentially important conformational off-state of a key regulatory multi-subunit protein kinase, SMG1. The study was enabled by applying a small molecule ATP-site inhibitor to capture the structure. The work will be of wide interest to the signaling and structural biology communities.

---

## [Decision Letter]

**Decision letter after peer review:**

Thank you for submitting your article "Cryo-EM reconstructions of inhibitor-bound SMG1 kinase reveal an autoinhibitory state dependent on SMG8" for consideration by *eLife*. Your article has been reviewed by 3 peer reviewers, and the evaluation has been overseen by Reviewing/Senior Editor (Philip Cole). The following individual involved in review of your submission has agreed to reveal their identity: Kacper B Rogala (Reviewer #1).

The Reviewing Editor has drafted this to help you prepare a revised submission. Although there are a substantial number of points listed below, the vast majority relate to manuscript presentation and writing issues rather than requested additional experimentation.

Essential revisions:

1. A significant concern with the manuscript is that there is no clear way to test the autoinhibition model. UPF1 binding itself is proposed to relieve autoinhibition, meaning that activity towards UPF1 would not be a readout of autoinhibition. One possible prediction of the model would be that activity of the complex toward a short peptide including the UPF1 phosphorylation sites should be low due to occlusion of the SMG1 catalytic cleft. Addition of an UPF1 truncation mutant lacking the C-terminal tail would be expected to activate the SMG1 complex toward phosphorylation of a peptide substrate. The SMG1-SMG9 dimer is modestly more active on UPF1 than the full SMG1-SMG8-SMG9 heterotrimer (see the Deniaud et al. paper cited in the text). However, using a peptide substrate, that difference in activity would be expected to us to be much larger. It would make the model more powerful if such ancillary enzymatic evidence were included.

2. Figure 1. Phosphorylation assays presented in this figure are missing an important control. And that is the reaction without the enzymes – for both SMG1 and mTOR. It is difficult to judge the extent of SMG1i inhibition if we cannot see the baseline with and without the enzymes. In general, monitoring of the AKT1 peptide phosphorylation with this method seems rather noisy, and this reviewer is unsure how relevant that specificity data is in the absence of proper controls. The same method presented in the authors' earlier paper (Gat et al., 2019, PMID: 31792449) seemed to be of higher quality.

3. Figure 1 is presented as providing evidence that SMG1i is selective for SMG1 over mTOR, and that it works by an ATP competitive mechanism. None of the experiments in Figure 1 or the associated supplementary figures actually investigate whether the compound is ATP-competitive; this was established later by the cryo-EM structure. It is worth noting that a non-selective analog bound at the ATP binding site when co-crystallized with PI3K-γ (reported in the paper that described SMG1i). Furthermore, in that original work, SMG1i was reported to be >400-fold selective for SMG1 over mTOR, and it was also tested against a number of other kinases. From this standpoint Figure 1 should be acknowledged as being confirmatory to that prior work rather than a new result, and it could be moved to a supplemental figure. There are also some technical issues with those experiments – the mTOR kinase assays used for replicates cover only a narrow concentration range and have high background; it's not clear why the full dose response shown in Figure 1 S2A wasn't simply repeated three times. The SMG1 activity assays were also performed at high kinase concentration, so the true potency of the compound cannot be determined. The original paper reported sub-nM inhibition of SMG1.

4. PRD density in the apo vs inhibitor/AMPPNP-bound mode of the SMG1-8-9 complex. The authors do not seem to offer any visual cues or thoughts to better understand why the PRD density would only rigidify in the presence of the inhibitor or ATP analogs. And within that group – why would the PRD density be more pronounced in the presence of the inhibitor versus the ATP analog? Are there any conformational changes in the SMG1-8-9 complex upon binding of the inhibitor/AMPPNP? Can the authors trace any specific residue/domain movement and rationalize this observation?

5. Methods. For the references cited in the methods, please use the primary articles on which the technique is developed rather than a subsequent paper that cites the original reference.

6. The presented structural data is clearly of high quality but additional figure panels covering the process of cryo-EM data processing in greater detail should be produced:

a. Representative micrograph of collected cryo-EM data.

b. 2D classes of their cryo-EM data.

c. Angular distribution plot for each deposited reconstruction in addition to their 3D FSC plot.

d. FSC plots for map vs. model.

7. Crosslinking-MS analysis on the SMG1-SMG8-SMG9 complex (as well as the complex including UPF1) was also previously done, and provided largely similar results to those reported in Figure 4B. Though not the intention of the authors, the use of bold lines for the highlighted crosslinks suggests that those specific interactions were identified at higher confidence or provided a higher signal; just keeping with the color scheme would be sufficient for the purpose of drawing attention to those crosslinks. In addition to the groove that approaches the IP6 binding site where additional density is seen in the inhibitor bound complex, the C-terminus of the insert region also crosslinks at multiple points within the catalytic domain itself. Can the authors reconcile the observed crosslinks to both regions?*Reviewer #1 (Recommendations for the authors):*

Claims/Observations:

(1) The SMG1i compound is specific to SMG1 because it engages unique residues in the active site (unique to SMG1 versus other PIKKs). Beyond the structural rationale provided by solid cryo-EM work, the authors attempted to strengthen this claim with in vitro kinase assays.

This reviewer believes that this data is of high value, but some controls for the in vitro kinase assays are missing and should be addressed to support this claim. Please see the detailed comments section.

(2) Structural rationale for the autoinhibitory function of the insertion domain of SMG1 and the C-terminal section of SMG8 – on the overall activity of the SMG1-8-9 complex.

Putative density belonging to the insertion domain (specifically its N-terminal section) is found in the active site of SMG1 – occluding the previously-mapped SQ substrate binding site, and explaining the previously-reported autoinhibitory function of this domain. Corresponding density has been observed in other PIKKs, and is normally referred to as PRD for PIKK regulatory domain. The authors were unable to unambiguously assign the amino-acid register for the putative PRD density due to its rather weak resolution. Their cross-linking mass spectrometry work confirmed that the insertion domain makes many contacts with the FATKIN domain of SMG1, and an extra few cross-links with the C-terminus of SMG8. in vitro binding assays with purified SMG1-insertion-domain and SMG8-C-terminus confirmed direct binding between these two regions of the complex, offering an explanation as to why the PRD section of the insertion domain is only rigidified in the context of the full SMG1-8-9 complex, and not the sub-complex lacking SMG8. They conclude that SMG8 stabilizes the PRD inside the SMG1 substrate binding site.

Given the wide range of methods applied to support this claim, this reviewer finds the evidence rather compelling. There are some points that should be addressed – either to clarify some statements or to expand on them. Please see the detailed comments section.*Reviewer #2 (Recommendations for the authors):*

Langer et al., present cryo-EM reconstructions of the PIKK family SMG1-9 and SMG1-8-9 kinase complexes bound to a SMG1 inhibitor or to an ATP analogue at sub-4 Å resolutions. Together, these structures show (1) the molecular interactions that define the specificity of a SMG inhibitor and (2) reveal an autoinhibitory function of the SMG1 insertion domain. Additional biochemical and cross-linking mass spectrometry work shows interactions between the SMG1 insertion domain and the SMG8 C-terminus. It clarifies how the SMG1 insertion domain and the SMG8 mediate autoinhibition of SMG1. Additionally, the authors show that the employed inhibitor has a high specificity for the inhibition of SMG1 in vitro.

The manuscript is written very clearly. The authors' data is overall well-presented and represents the first structure of SMG1 bound to an inhibitor. The presented structural data is of high quality and appears plausible. Together, the authors' claims are supported by the presented data but additional figures and expansion of the methods section to present their cryo-EM data analysis in greater detail will increase overall clarity.

*Reviewer #3 (Recommendations for the authors):*

The SMG1-SMG8-SMG9 kinase complex phosphorylates the protein UPF1 to promote nonsense-mediated mRNA decay, an essential co-translational quality control mechanism for degradation of messages with premature stop codons. This manuscript from Langer et al. describes cryo-EM structures of the SMG1-SMG8-SMG9 complex bound to a selective ATP-competitive small molecule inhibitor. The structure provides a rationale for why the compound is selective for SMG1 over related kinases such as mTOR and DNA-PK, which should prove useful in furthering inhibitor design and development for this class of kinases. Intriguingly, the inhibitor bound structure revealed additional density absent from prior reconstructions of the complex arising from am "SMG1 insertion region" C-terminal to its catalytic domain. While this density could not be fit to an atomic model, a portion of it occludes the catalytic cleft, suggesting that it mediates cis-autoinhibitory regulation of the kinase. Furthermore, this region appears to contact the SMG8 subunit, and the added density is absent from cryo-EM reconstructions of an SMG1-SMG9 complex lacking SMG8. Both the SMG1 intramolecular contacts and SMG1-SMG8 intermolecular contacts were supported by crosslinking-mass spectrometry analysis, which appeared largely similar to a previously reported analysis using the same method. These observations support a model by which SMG8, through direct interactions with the SMG1 insertion region, has a key role in autoinhibition of the kinase. These studies are important for the field in that how the SMG1 complex is regulated has previously been obscure. A weakness of the study is this structural hypothesis could not be independently tested. Indeed, given that the SMG1 insertion region was not observed at high resolution, it would not be possible to design mutants to disrupt key interactions and examine their impact on SMG1 kinase activity. The authors hypothesize that these autoinhibitory interactions are relieved by substrate binding, which is consistent with prior cryo-EM characterization of the SMG1-SMG8-SMG9-UPF1 complex. However, this potential substrate activation mechanism, while plausible, is not directly addressed in the manuscript.

---

## [Author Response]

Essential revisions:1. A significant concern with the manuscript is that there is no clear way to test the autoinhibition model. UPF1 binding itself is proposed to relieve autoinhibition, meaning that activity towards UPF1 would not be a readout of autoinhibition. One possible prediction of the model would be that activity of the complex toward a short peptide including the UPF1 phosphorylation sites should be low due to occlusion of the SMG1 catalytic cleft. Addition of an UPF1 truncation mutant lacking the C-terminal tail would be expected to activate the SMG1 complex toward phosphorylation of a peptide substrate. The SMG1-SMG9 dimer is modestly more active on UPF1 than the full SMG1-SMG8-SMG9 heterotrimer (see the Deniaud et al. paper cited in the text). However, using a peptide substrate, that difference in activity would be expected to us to be much larger. It would make the model more powerful if such ancillary enzymatic evidence were included.

This point addresses two different albeit connected aspects: autoinhibition and activation.

The auto-inhibition mechanism is the focus of this manuscript.

There have been several papers suggesting the presence of negative regulation based on biochemical data but the molecular mechanism for these observation had remained unclear. For example, a negative regulatory effect of the SMG1 insertion domain has been reported in Deniaud et al., 2015 and in Zhu et al., 2019. Our observation from the cryo-EM analyses that the SMG1 insertion domain directly blocks the substrate binding path within the SMG1 active site rationalizes the negative regulation observed in biochemical data. In addition, autoinhibition by the PRD/insertion domain has been observed structurally in related kinases (see for example Stakyte et al., 2021 (human ATM) or Chen et al., 2020 (human DNA-PKc)). Furthermore, Arias-Palomo et al., 2011, Deniaud et al., 2015 and Zhu at al., 2019 described a negative regulatory effect of SMG8. Our structural data rationalize this effect: e.g. the dependency of SMG1 autoinhibition on the presence of SMG8 in the complex, the observation of close proximity of the SMG1 insertion domain and the SMG8 C-terminus as well as our discovery of a direct interaction between these parts of the complex via pull-downs using purified proteins. Overall, this allows us to integrate the existing biochemical data into a model for the concerted action of the SMG1 insertion domain and the SMG8 C-terminus to induce an autoinhibited state of the SMG1-8-9 kinase complex.

The activation mechanism is a very interesting aspect and we agree with the Reviewer/s that this remains unclear. However, the activation mechanism is not the focus of this manuscript. We briefly discuss the possible relationship between autoinhibition and activation at the end of the manuscript, to point out unanswered questions and future avenues of research.

We are grateful for the suggestions of the Reviewer/s on how to tackle the activation aspect of the process, though there are technical issues that complicate such experiments.

1) Our data indicate that ideal phosphorylation motifs bind the SMG1 active site more tightly than the SMG1 insertion domain (high resolution observed for substrate peptide in Langer et al., 2020, as compared to medium resolution observed for SMG1 insertion domain in the current manuscript). Therefore, depending on its concentration relative to the kinase, an ideal peptide will be able to compete with the insertion domain autoinhibition in this in vitro setup without any additional activation. Higher peptide concentrations might simply mimic the effect of optimal positioning of the UPF1 N- and C-terminal regions with respect to the SMG1 kinase in an in vivo NMD complex, independent of direct interaction of SMG1-8-9 and UPF1. We have included an additional sentence to our discussion to stress this possibility:

"An alternative explanation would be that optimal positioning of UPF1 with respect to the SMG1 kinase in vivo will allow the unstructured N- and C-terminal ends of UPF1 to efficiently compete with the SMG1 insertion domain for binding to the SMG1 active site. This would be mimicked by the high protein concentration used in in vitro assays (such as Figure 4 Supp. 3 B)".

2) Our SMG1-8-9 purifications contain a mixture of SMG1-8-9 and SMG1-9 complexes, as evidenced by the reconstructions reported in the current manuscript and we were so far not able to separate these subpopulations during purification. The presence of a significant amount of SMG1-9 in our purifications means that any effect observable in the assay suggested by the Reviewer/s would be greatly diluted. Further aggravating this issue, overexpression of only SMG1 and SMG9, but not SMG8, in HEK293T cells results in preparations that still contain endogenous SMG8 (see Author response image 1).

**Author response image 1. sa2fig1:** SDS-PAGE analysis of complexes purified from cells transfected to stably express either TwinStrep-SMG1-8-9 or TwinStrep-SMG1-9, as indicated. Note the co-purification of endogenous SMG8 from the cell line not transfected with SMG8.

In general, understanding of how SMG1-8-9 activation is regulated will require insights into the molecular interplay between SMG1-8-9, UPF1, other NMD factors and a terminating ribosome. This is an area of future studies that goes beyond the scope of this manuscript.

2. Figure 1. Phosphorylation assays presented in this figure are missing an important control. And that is the reaction without the enzymes – for both SMG1 and mTOR. It is difficult to judge the extent of SMG1i inhibition if we cannot see the baseline with and without the enzymes. In general, monitoring of the AKT1 peptide phosphorylation with this method seems rather noisy, and this reviewer is unsure how relevant that specificity data is in the absence of proper controls. The same method presented in the authors' earlier paper (Gat et al., 2019, PMID: 31792449) seemed to be of higher quality.

We thank the Reviewer/s for pointing this out. We have added three new experiments to address this point:

We have included assays with a "no kinase" control for both SMG1 and mTOR (Figure 1 Supp. 2 D and E).

As a further control, we have included the mTOR inhibitor Torin 2 into this experiment. This assay was carried out in triplicates for both SMG1 as well as mTOR allowing quantification (Figure 1 Supp. 2 F).

In addition, we have also repeated the mTOR triplicate experiment shown in Figure 1 E and Figure 1 Supp. 2 C to improve the quality of the gels. We would also like to draw the attention of the reviewer to the Figure captions and the Methods section, where we describe that we quantify mTOR autophosphorylation to avoid problems with the weaker AKT1 peptide signal.

3. Figure 1 is presented as providing evidence that SMG1i is selective for SMG1 over mTOR, and that it works by an ATP competitive mechanism. None of the experiments in Figure 1 or the associated supplementary figures actually investigate whether the compound is ATP-competitive; this was established later by the cryo-EM structure.

We agree with the Reviewer/s. We have removed claims concerning ATP competition in context with Figure 1 and introduce this observation when describing the structural data.

It is worth noting that a non-selective analog bound at the ATP binding site when co-crystallized with PI3K-γ (reported in the paper that described SMG1i). Furthermore, in that original work, SMG1i was reported to be >400-fold selective for SMG1 over mTOR, and it was also tested against a number of other kinases. From this standpoint Figure 1 should be acknowledged as being confirmatory to that prior work rather than a new result, and it could be moved to a supplemental figure.

We agree with the Reviewer/s that the data on SMG1**i** compound presented in Figure 1 are confirmatory, although we point out that this is actually the first time that in vitro raw data using purified components are shown (in Gopalsamy et al., 2012, there is a Table reporting affinities, but there are no in vitro raw data nor gels showing the quality of enzymes used for the characterization). The data on SMG1**i** compound presented in Figure 1 are meaningful in the context of this manuscript, as they provide the biochemical evidence for the feasibility of structural characterization using this experimental system. Since these data set the stage for the structural analysis, we would prefer to keep these data in main figure, while specifying that they confirm/validate previous reports:

"In agreement with previous results (Gopalsamy et al. 2012), we concluded that SMG1i displays high potency and specificity in inhibiting the SMG1 kinase in vitro."

There are also some technical issues with those experiments – the mTOR kinase assays used for replicates cover only a narrow concentration range and have high background; it's not clear why the full dose response shown in Figure 1 S2A wasn't simply repeated three times. The SMG1 activity assays were also performed at high kinase concentration, so the true potency of the compound cannot be determined. The original paper reported sub-nM inhibition of SMG1.

As detailed in our response to point 2, we have repeated the mTOR triplicate experiment and now provide higher quality gels with less background in Figure 1 E and Figure 1 Supp. 2 C. The goal of this experiment was to confirm the specificity of SMG1**i** for SMG1 over mTOR qualitatively in our experimental setup before proceeding to structural analysis. The concentration range tested in triplicates was chosen based on the titration experiments shown in Figure 1 C and Figure 1 Supp. 2 A. It is beyond the scope of this study to provide a comprehensive characterization of the pharmacological properties of SMG1i.

4. PRD density in the apo vs inhibitor/AMPPNP-bound mode of the SMG1-8-9 complex. The authors do not seem to offer any visual cues or thoughts to better understand why the PRD density would only rigidify in the presence of the inhibitor or ATP analogs. And within that group – why would the PRD density be more pronounced in the presence of the inhibitor versus the ATP analog? Are there any conformational changes in the SMG1-8-9 complex upon binding of the inhibitor/AMPPNP? Can the authors trace any specific residue/domain movement and rationalize this observation?

Overall, we observed better defined density for the entire kinase active site when either a nucleotide analogue or the inhibitor is present as compared to our previously published apo-structure (Gat et al., 2019). We believe that the extended interface between the inhibitor and SMG1 reaching closer towards the PRD (as compared to AMPPNP) might further contribute to stabilization (Figure 2 Supp. 3 D).

We did not detect conformational changes upon ligand binding to the active site. Due to the limited resolution of the PRD density, we cannot interpret interactions at the residue level.

5. Methods. For the references cited in the methods, please use the primary articles on which the technique is developed rather than a subsequent paper that cites the original reference.

We included additional citations where we noticed we had not cited the primary articles on which the technique is developed, such for Relion and Coot for example.

6. The presented structural data is clearly of high quality but additional figure panels covering the process of cryo-EM data processing in greater detail should be produced:a. Representative micrograph of collected cryo-EM data.b. 2D classes of their cryo-EM data.c. Angular distribution plot for each deposited reconstruction in addition to their 3D FSC plot.d. FSC plots for map vs. model.

We have now split Figure 2 Supp. 1 into two Figures (Figure 2 Supp.1 and Figure 3 Supp. 2) and included map vs. model FSC plots as well as angular distribution plots for each reconstruction. In addition, we included 2D class averages calculated from SMG1-8-9 and SMG1-9 particle stacks in Figure 2 Supp. 2 A to illustrate differences between the complexes in two dimensions. We did not include a micrograph or 2D class averages originating from initial classifications as these have been shown before for our SMG1-8-9 preparation (Gat et al., 2019, Langer et al., 2020). We point out that this manuscript was indeed written as a Research Advance to be connected to Langer et al., 2020.

7. Crosslinking-MS analysis on the SMG1-SMG8-SMG9 complex (as well as the complex including UPF1) was also previously done, and provided largely similar results to those reported in Figure 4B.

There are several reasons why we decided to carry out an independent XL-MS analysis:

1) to the best of our knowledge, raw data of the previously published cross-linking MS experiments (Deniaud et al., 2015) are not publicly available, and we therefore decided to obtain raw data ourselves.

2) We wanted to carry out cross-linking experiments in buffer conditions identical to the ones we used for structure determination (for the best possible correlation).

We will upload a spreadsheet with the identified cross-links and other relevant data to accompany the manuscript.

In the revised manuscript, we could actually go a step further, as AlphaFold (Jumper et al. 2021) became available and predicts the fold for the SMG8 C-terminal domain. The AlphaFold structural prediction of the SMG8 C-terminal domain fits remarkably well in the features of the cryo-EM density and allows to map crosslinks we had detected with this portion of the molecule. Thus, our XL-MS and EM data, in combination with the AlphaFold prediction allows us to provide a more complete model of the SMG1-8-9 complex, as illustrated in the newly added Figure 4 Supp. 3.

Though not the intention of the authors, the use of bold lines for the highlighted crosslinks suggests that those specific interactions were identified at higher confidence or provided a higher signal; just keeping with the color scheme would be sufficient for the purpose of drawing attention to those crosslinks.

We have changed the display in Figure 4 B and do now highlight cross-links only by color to avoid misinterpretation.

In addition to the groove that approaches the IP6 binding site where additional density is seen in the inhibitor bound complex, the C-terminus of the insert region also crosslinks at multiple points within the catalytic domain itself. Can the authors reconcile the observed crosslinks to both regions?

There are indeed more cross-links that support our assignment of the additional density to the SMG1 insertion C-terminus region: K2363 and K2370 of the SMG1 kinase domain cross-link to multiple residues of the SMG1 insertion C-terminus and are in close proximity to the observed additional density, albeit located in regions closer to the C-terminal end of the insertion domain (see Author response image 2). We chose to highlight the residues shown in Figure 4 C since this is where the additional density is the strongest and it allows us to illustrate the proximity to the IP6 binding site.

**Author response image 2. sa2fig2:** Residues of the SMG1 kinase domain (K2363, K2370) that cross-link to the SMG1 insertion C-terminus are observed in close proximity to the additional density (red). Related to Figure 4 C.

There are two residues within the SMG1 kinase site that also cross-link to the SMG1 insertion C-terminus, but are not located in direct proximity to the observed additional density. We think that this reflects the generally flexible nature of the SMG1 insertion domain, which is also indicated by the low-resolution of the visualized densities.